# Insights into the Role of Fungi in Pine Wilt Disease

**DOI:** 10.3390/jof7090780

**Published:** 2021-09-20

**Authors:** Cláudia S. L. Vicente, Miguel Soares, Jorge M. S. Faria, Ana P. Ramos, Maria L. Inácio

**Affiliations:** 1Mediterranean Institute for Agriculture, Environment and Development (MED), Institute for Advanced Studies and Research, Universidade de Évora, 7006-554 Évora, Portugal; jorge.faria@iniav.pt; 2Instituto Nacional de Investigação Agrária e Veterinária (INIAV, I.P.), 2780-159 Oeiras, Portugal; 3Laboratório de Patologia Vegetal “Veríssimo de Almeida” (LPVVA), Instituto Superior de Agronomia (ISA), University of Lisbon, 1349-017 Lisboa, Portugal; soares8118@gmail.com (M.S.); pramos@isa.ulisboa.pt (A.P.R.); 4Linking Environment Agriculture and Food (LEAF), Instituto Superior de Agronomia (ISA), University of Lisbon, 1349-017 Lisboa, Portugal; 5GREEN-IT Bioresources for Sustainability, Instituto de Tecnologia Química e Biológica, Universidade Nova de Lisboa (ITQB NOVA), Av. da República, 2780-157 Oeiras, Portugal

**Keywords:** biocontrol, blue-stain fungi, interactions, mycobiome, pine wood nematode

## Abstract

Pine wilt disease (PWD) is a complex disease that severely affects the biodiversity and economy of Eurasian coniferous forests. Three factors are described as the main elements of the disease: the pinewood nematode (PWN) *Bursaphelenchus xylophilus*, the insect-vector *Monochamus* spp., and the host tree, mainly *Pinus* spp. Nonetheless, other microbial interactors have also been considered. The study of mycoflora in PWD dates back the late seventies. Culturomic studies have revealed diverse fungal communities associated with all PWD key players, composed frequently of saprophytic fungi (i.e., *Aspergillus*, *Fusarium*, *Trichoderma*) but also of necrotrophic pathogens associated with bark beetles, such as ophiostomatoid or blue-stain fungi. In particular, the ophiostomatoid fungi often recovered from wilted pine trees or insect pupal chambers/tunnels, are considered crucial for nematode multiplication and distribution in the host tree. Naturally occurring mycoflora, reported as possible biocontrol agents of the nematode, are also discussed in this review. This review discloses the contrasting effects of fungal communities in PWD and highlights promising fungal species as sources of PWD biocontrol in the framework of sustainable pest management actions.

## 1. Introduction

Pine wilt disease (PWD) has become one of the most damaging diseases to conifers worldwide and is a risk nowadays for the sustainability and profitability of forest ecosystems. PWD was detected in Japan in 1905, later spreading to other Asian countries; namely China, Korea, and Taiwan in the 1980s [1], and to Europe (Portugal and Spain in 1999 and 2010, respectively) [2,3,4]. To date, no recorded outbreaks have been identified in other European countries, despite several scientific studies alert for a high vulnerability of northern European pine forests due to the oncoming effects of climate change and the reported susceptibility of the dominating pine species to PWD [5,6]. The PWD has gradually and consistently spread between continents as a result of an increase in the global trade of wood and derivative materials. Significant economic and ecological impacts were reported in the affected countries, including a reduction in productivity and an increase in the costs of management procedures for disease control, as well as a decrease in forest biodiversity [7,8].

Effective PWD management strategies have been difficult to achieve given the complex disease infection cycle, where several organisms contribute to the infection’s overall development and severity, namely its causal agent, the pinewood nematode (PWN) *Bursaphelenchus xylophilus* (Steiner & Bührer) Nickle; the PWN’s insect-vector *Monochamus* spp.; and a susceptible host tree, commonly trees from the genus *Pinus* [1]. Recent studies have additionally identified a strong influence of the PWN-associated bacteria, and the microbiome associated with the susceptible pine, on PWD development [9,10,11]. While the description of microbial diversity has been established for the PWN, the insect-vector, and susceptible pine hosts—leading to the preliminary proposal of their functional roles in the PWD—information on the mycological diversity associated with these organisms is scarcer. In the present review, an up-to-date compilation of the published works reporting on naturally occurring fungi associated with the PWN, the host tree, and its insect-vector, is presented. A critical analysis of the summarized information further allows us to envision the functional role of associated fungi on PWD development and guide future research in this area.

## 2. The Complexity of PWD

The PWN is a small plant-parasitic nematode of about 500–1000 µm in length and 22 µm in width, capable of feeding on plant (phytophagous) and fungal tissues (mycophagous). Its life cycle is comprised of four juvenile stages (J1 to J4) and both adult males and females (Figure 1a). In natural conditions, the PWN can quickly complete its life cycle, usually 4 days in summer conditions while, under in vitro conditions in laboratory cultured *Botrytis cinerea* Pers. mats, it can take between 4 to 6 days [1,12] (or slightly longer on aseptic co-cultures of in vitro pine with the PWN) [13]. Adverse environmental conditions and/or undernutrition induce morphological and physiological changes in the J3 stage, leading to the occurrence of the PWN dispersal stages, the third and fourth dispersal juvenile stages (J_III_ and J_IV_). The J_IV_, known as the dauer juvenile stage, is characterized by an interruption of the feeding process, the establishment of large lipidic reserves, and the production of a thick protective layer around the PWN’s body [1,14]. At this stage, the PWN is attracted to the juvenile longhorn beetles emerging from dead, or decaying, wood in forest ecosystems, invading beetles trachea. The PWN establishes these commensalism associations with members of the genus *Monochamus* (Coleoptera; Cerambycidae), considered the main PWN vector, with seven confirmed species vectoring the nematode in field and laboratory conditions [15,16,17], the most common being *M. alternatus* in East Asian forestlands [1], and *M. galloprovincialis* in Europe [18,19]. Colonized beetles can transmit the PWN by transporting them to (1) new beetle feeding sites during the beetle maturation stage, commonly in young branches of susceptible pine species (primary transmission), or (2) weakened or dead trees in the forest, where the matured female beetles oviposit (secondary transmission). The beetles benefit from an increase in reproduction sites at weakened or dead trees, where oviposition occurs, while the PWN takes advantage of the increase in beetle population, leading to higher rates of transmission (Figure 1b) [20]. Once infection occurs in a susceptible tree, the PWN quickly multiplies and begins feeding on the epithelial parenchyma cells lining the pine resin ducts, inducing extensive damage that leads to a reduction in resin production and the release of volatile terpenoids. As the infection progresses, the PWNs damage the tree’s vascular system and an embolism phenomenon begins to interrupt water transport, partly due to the build-up of volatile terpenoids [20,21]. The typical symptoms of PWD (Figure 2) are visible at this stage, namely pine shoot wilting due to desiccation, chlorosis, and drooping. When external symptoms of the PWD are noticeable, the affected tree cannot recover and eventually dies. However, the symptoms caused by the PWN, including abrupt reduction in the production of resin and pine needle wilting, can be mistakenly attributed to other biotic or abiotic stress factors such as water stress [22]. In North America, endemic pines are mostly tolerant, with only exotic species expressing the strongest symptoms. In China, Japan, Korea, and Portugal, many pine species were found to be very susceptible, e.g., *Pinus densiflora* Sieblod & Zucc., *Pinus nigra* J.F. Arnold, *Pinus pinaster* Aiton, *Pinus radiata* D. Don, *Pinus sylvestris* L. or *Pinus thunbergii* Parl., which caused extensive economic and cultural repercussions [15,23].

Despite not being able to induce PWD, the microbiome (bacteria and fungi) associated with the PWD, and its elements is considered a major biotic factor influencing the disease’s severity (further detailed in [9,10,11]). The isolation of fungi and bacteria dates back the 70s, when Tokushige and Kiyohara isolated microbial samples from dead pine trees and tested their pathogenicity in healthy pines [20]. Since then, several researchers attempted to correlate the pathogenicity of some bacterial species, like *Pseudomonas* spp. or *Bacillus* spp., with the disease [24,25,26]. Still, the latest studies point to a less active role of the bacterial communities associated with PWD. These communities behave as opportunistic/saprophytes and/or endophytes expressing phenotypic plasticity in the PWN-wilted pine host [11,27,28,29,30,31]. The impact of fungal communities in the complexity of PWD and their elements is presented in detail in the next section.

## 3. Mycobiome Associated with PWD Complex

Forest trees harbour extremely complex fungal communities that play important roles on ecosystem multifunctionality and equilibrium [32]. Most of these communities are shaped by intrinsic factors of the host, like host genotype, condition, and/or development, and by external factors such as geographical location, seasonality, or even surrounding vegetation [33]. Under the presence of non-native pathogens, these ecosystems suffer disequilibrium situations that can eventually lead to considerable spatial and temporal community variation [34].

### 3.1. Diversity and Composition of Fungal Communities

Fungal communities associated with PWD have been described since the early 80s. These culture-dependent studies worked with different culture conditions, such as different growth media (e.g., potato dextrose agar or malt extract agar) or supplementation of antibiotics (Table 1). The most recent reports in fungal communities associated with the disease combine morphology with molecular identification based on fungal DNA barcode markers (i.e., primary marker ITS, internal transcribed spacer; and secondary marker TEF1-α, translational elongation factor 1 alpha) [35,36] and other protein-coding genes (i.e., beta-tubulin BT or calmodulin CAL) [37]. These conventional culturing methods are biased towards fast-growing species rather than the more specialized fungi [38], suggesting that only a limited fraction of the fungal community’s diversity has been uncovered in this complex disease. Three main phyla were identified within the Fungi kingdom, namely Ascomycota, Basidiomycota, and Mucoromycota. The most predominant phylum, Ascomycota, was represented by 6 classes (Blastomycetes, Dothideomycetes, Eurotiomycetes, Leotiomycetes, Orbiliomycetes, and Sordariomycetes) and nearly 30 families (Table 1). Most of the described taxa are common saprophytes and probably not specific to the disease or associated with the PWN. This may be the case of *Penicillium* (Ascomycota, Eurotiomycetes, Trichocomaceae), *Trichoderma* (Ascomycota, Sordariomycetes, Hypocreaceae) and *Aspergillus* (Ascomycota, Eurotiomycetes, Aspergillaceae) among others, which are ubiquitous to all existing environments and detected in all PWD elements (Table 1). The description of fungal communities has also been made in different pine species and insect-vectors from different geographical locations (e.g., China, Japan, Korea, Portugal, and the USA). The first isolations of mycoflora were conducted on symptomatic *P. thunbergii* (shoots, twigs, and woodchips), on the surface tissues of tunnels and pupal chambers bored by *Monochamus* larvae, and from the adult body of *M. alternatus* after its emergence [39,40]. These works reported that the genera *Ceratocystis* and *Verticicladiella* (synonym of *Leptographium* [41]), from Ceratocystidaceae and Ophiostomataceae families, respectively, were the only flora common to the three sampled locations apart from the saprophytic fungi *Trichoderma* and *Penicillium*. Later, Wingfield [42] isolated fungi from the cerambycid beetles *M. scutellatus* and *M. carolinenses*, also identifying *Ceratocystis* and *Ceratocystiopis* as common genera associated with adult beetles and pupal chambers from *P. banksiana* and *P. resinosa*. Curiously, Wingfield [42] could also isolate the nematode-trapping fungi *Arthrobotrys cladodes* var. *cladodes* and *A. superba* in the PWN. Kuroda and Iko [43] isolated fungi from healthy and wilted *P. thunbergii* and reported that the same species were recovered from both pine trees (i.e., *Pestalotiopsis* spp., *Nigrospora* spp., *Cladosporium* spp., and *Phomopsis* spp.) and that *Ceratocystis* sp., was only detected after PWN inoculation. This study has also shown that the composition of fungal species varied slightly among seasons [43]. In 2007, Hyun et al. [44] characterized the fungal communities associated with the PWN; the insect-*vector M. alternatus* and *P. thunbergii* in Korea. Among the 15 genera identified, *Penicillium* and *Ophiostoma* were the most frequent genera in all elements, with PWNs and insect larvae showing a smaller number of associated fungi than insect adults or infected wood. In 2015, Inácio et al. [45] described fungal communities associated with *M. galloprovincialis*, native to Portugal. From a total of 100 insects, species of 18 genera of filamentous fungi were reported: *Acremonium*, *Alternaria*, *Arthrinium*, *Aspergillus*, *Beauveria*, *Bipolaris*, *Botryosphaeria*, *Botrytis*, *Cladosporium*, *Clonostachys*, *Eppicocum*, *Fusarium*, *Ophiostoma* s. l., *Paecilomyces* s. l., *Penicillium*, *Stemphylium*, *Trichoderma*, and *Trichothecium* (Table 1) [45]. A more detailed characterization of endophytic fungi associated with *P. pinaster* with, and without, PWN infection in Portugal was presented by Trindade [46]. Novel fungal species are also continuously described. Wang et al. [47] identified new species of Ophiostomatales associated with PWD in the pupal chambers of *M. alternatus* from infected *Pinus massoniana* Siebold & Zucc. and *P. thunbergii* in China. In this study, over 90% of all isolates were identified as *Ophiostoma ips*, with three novel species—*Ophiostoma album* sp. nov., *Ophiostoma massoniana* sp. nov. and *Sporothrix zhejiangensis* sp. nov.—and two species whose identities remained unclear; *Ophiostoma* cf. *deltoideosporum* and *Graphilbum* cf. *rectangulosporium* [47].

The availability of high-throughput sequencing (HTS) technologies, such as amplicon metagenomics, has revolutionized ecological studies of fungal communities [37] allowing broader insight into the complexity of host-fungal interactions. The primary fungal DNA barcode ITS is commonly used on HTS-based metabarcoding. Still, caution should be taken when interpreting HTS results since, for several groups of important plant pathogens and endophytes, ITS provides insufficient resolution for species-level assignment [37]. It is recommended that researchers use the ITS2 subregion, less taxonomically biased, with lower length variation and more universal primer sites, or the full ITS region with greater taxonomic resolution and reduced amplification of dead organisms [37]. In the context of PWD, amplicon technologies were firstly applied to characterize bacterial communities (e.g., using the hypervariable regions of 16S rRNA molecule), revealing the presence of PWNs and PWD progression [48,49,50]. Chu et al. [51] described the impact of the disease on root-associated fungi (e.g., ectomycorrhizal fungi, ECMF; dark septate endophytic fungi, DSE; arbuscular mycorrhizal fungi, AMF) in different stands of *Pinus tabulaeformis* Carriére forest (undisturbed, moderate, and highly disturbed pine stands). The authors showed that fungal community richness and diversity, as well as soil hyphal density, decreased with the increase of disease disturbance. Basidiomycota and Ascomycota were the dominant root-associated fungi, with specific genera present in the different disturbed stands [51].

**Table 1 jof-07-00780-t001:** Culture-dependent mycoflora isolated from the PWN, insect-vectors, and host pines. PWN, *Bursaphelenchus xylophilus*; Ma, *Monochamus alternatus*; Ms, *M. scutellatus*; Mg, *M. galloprovincialis* (A—Adult, L—Larvae); Pd, *Pinus densiflora*; Pm, *P. massoniana*; Pt, *P. thunbergii*; Pb, *P. banksiana*; Pr, *P. resinosa*; Pp, *P. pinaster* (W—wood, Pc, pupal chamber); JP, Japan; PT, Portugal; KR, Korea; CH, China; USA, United States of America.

Fungal Taxonomy					Insect-Vector						Host Pine								Country	References
					Ma		Ms		Mc	Mg	Pd		Pm	Pt		Pb	Pr	Pp		
Phylum	Class	Family	Genus	PWN	A	L	A	L	A	A	W	Pc	Pc	W	Pc	Pc	Pc	W		
Ascomycota	Blastomycetes	Crytococcaceae	*Candida*								•								JP	[52]
	Dothideomycetes	Botryosphaeriaceae	*Diplodia*								•			•					JP	[39,52]
			*Botryosphaeria*							•									PT	[46]
			*Macrophoma*								•								JP	[53]
			*Sphaeropsis*											•					JP	[54]
		Cladosporiaceae	*Cladosporium*		•					•	•			•				•	JP, PT	[39,43,45,46,52]
		Didymellaceae	*Epicoccum*							•								•	PT	[45,46]
		Dothideaceae	*Rhizosphaera*								•									[39]
		Leptosphaeriaceae	*Leptosphaeria*		•														KR	[44]
		Massarinaceae	*Helminthosporium*				•	•											USA	[42]
		Pleosporaceae	*Alternaria*	•	•		•	•		•	•							•	JP, PT	[39,45,46,52]
			*Bipolaris*							•									PT	[45]
			*Curvalaria*	•															CH	[55]
			*Stemphylium*					•											PT	[45]
		Saccotheciaceae	*Aureobasidium*				•	•		•				•					JP, PT	[45,46,54]
	Eurotiomycetes	Aspergillaceae	*Aspergillus*	•	•		•	•		•	•			•				•	JP, CH, PT	[42,44,45,46,52,54,55]
		Trichocomaceae	*Penicillium*	•	•	•	•	•		•	•			•	•	•	•	•	JP, CH, PT	[39,44,45,46,52,53,54]
			*Paecilomyces*							•								•	PT	[45,46]
		Herpotrichiellaceae	*Phialophora*											•		•	•		JP	[42,52]
			*Rhinocladiella*													•	•		USA	[42]
	Leotiomycetes	Sclerotinuaceae	*Botrytis*							•					•			•	CH, PT	[44,45,46]
	Orbiliomycetes	Orbiliaceae	*Arthrobotrys*	•							•					•	•		JP, CH, USA	[42,52,53]
			*Dactylaria*								•								JP	[52]
	Sordariomycetes	Amphisphaeriaceae	*Pestalotia*		•														JP	[39]
		Aspiosporaceae	*Arthrinium*							•								•	PT	[45,46]
		Bionectriaceae	*Bionectria*												•				KR	[44]
			*Clonostachys*							•								•	PT	[45,46]
		Boliniaceae	*Camarops*												•				KR	[44]
		Ceratocystidaceae	*Ceratocystis*	•	•		•	•	•		•	•		•		•	•		JP, CH, USA	[39,42,44,53,55]
		Chaetomiaceae	*Chaetomium*	•															CH	[55]
		Cordycipitaceae	*Beauveria*				•	•		•								•	USA, PT	[42,45,46]
		Glomerellaceae	*Colletotrichum*	•	•		•	•											JP, CH	[39,55]
		Hypocreaceae	*Hypocrea*		•										•				KR	[44]
			*Cephalosporium*								•								JP	[52]
			*Gliocladium*											•		•	•		JP, USA	[42,54]
			*Trichoderma*	•			•	•		•	•			•	•	•	•	•	JP, CH, USA, PT	[39,42,44,45,46,52,53,54,55]
		Microascaceae	*Graphium*								•								JP	[52]
		Nectriaceae	*Gibberella*		•														KR	[44]
			*Fusarium*	•						•	•			•	•			•	JP, CH, KR, PT	[39,44,45,46,53,54,55]
			*Mariannaea*								•								JP	[52]
			*Nectria*												•				KR	[44]
		Ophiostomataceae	*Ceratocystiopsis*				•	•	•							•	•	•	USA, PT	[42,46]
			*Graphilbum*										•	•				•	KR, PT	[46,47,50]
			*Leptographium*								•								JP	[39,53]
			*Ophiostoma*	•	•	•				•			•	•	•			•	CH, KR, PT	[44,45,46,47,56]
			*Sporothrix*										•	•				•	CH, PT	[46,47]
		Plectosphaerellaceae	*Plectosphaerella*		•														KR	[44]
			*Verticillum*	•										•					JP	[53]
		Sordariaceae	*Sordaria*	•															CH	[55]
		Sporocadaceae	*Monochaetia*	•															CH	[55]
			*Pestalotiopsis*	•							•			•				•	JP, CH, PT	[43,46,52,53,54,55]
		Trichosphaeriaceae	*Nigrospora*	•										•					JP, CH	[43,55]
		Xenospadicoidaceae	*Spadicoides*												•	•			USA	[42]
		Valsaceae	*Phomopsis*	•							•			•					JP	[52,53,55]
		Incertae sedis	*Trichothecium*							•								•	PT	[45,46]
Basidiomycota	Agariomycetes	Irpicaceae	*Irpex*		•														KR	[44]
		Ceratobasidiaceae	*Rhizoctonia*											•					JP	[54]
Mucoromycota	Mortierellomycetes	Mortierellaceae	*Mortierella*											•					JP	[54]
	Mucoromycetes	Mucoraceae	*Mucor*	•										•	•				KR	[44]
		Rhizopodaceae	*Rhizopus*	•															CH	[55]

Later, the same authors assessed the effects of the PWN on the colonization rates, and community structure/diversity of root-associated mycoflora (EMCF, DSE, and AMF) in PWN-infected and non-infected *P. tabulaeformis* [57]. Once more, infection with the PWN reduced the biomass, abundance, and colonization of root-associated fungi, as well as species richness and diversity. Zhang et al. [50] analyzed the endophytic (stem and branches) and rhizosphere fungal communities in healthy (without PWNs) and PWN-infected *P. massoniana* using ITS1 (ITS1f-ITS2R) metagenomics. The study revealed that healthy pines (stem and branches) have a higher endophytic fungi species richness and diversity than wilted pines (stem/branches) and rhizospheres (from healthy and wilted pines) suggesting that PWN infections likely affect endophytic fungi communities. No significant differences were found between the fungal communities of wilted pines and those of rhizosphere fungi in healthy and wilted pines. Phylum Ascomycota had higher abundance (ca. 75%), especially on samples from wilted pines. The shared fungal communities between healthy and wilted *P. massoniana* were species of *Cyberlindnera*, *Kirschsteiniothelia*, *Penicillifer*, *Penicillium*, *Pestalotiopsis*, *Saitozyma*, *Sporothrix*, *Trichoderma*, *Venturia*, and *Zygoascus*. The abundances of *Penicillifer*, *Zygoascus*, *Kirschsteiniothelia*, and *Sporothrix* were higher in wilted, than in healthy, pines [50]. More recently, Liu and colleagues [58] analyzed the fungal community and functional structure in needles, roots, and surrounding soil of *P. thunbergii* naturally infected by PWNs, targeting full ITS (ITS1f-ITS2R). No significant changes on fungal diversity were found between the soil or roots of healthy and diseased trees, contradicting the previous studies [51,57]. Fungal species richness/diversity/evenness and community structure in the needles of diseased trees were significantly lower than those in healthy trees [58]. The most predominant phyla were Ascomycota (54.6% of the operational taxonomy units, OTUs), followed by Basidiomycota (13.8% OTUs) and Mortierellomycota (1.8% OTUs) and, at genus level, *Mortierella* (Mortierellomycota), *Delicatula* (Basidiomycota), and *Trichoderma* (Ascomycota) were the most abundant. In terms of functional prediction, saprotrophs were identified in higher abundance in needles of PWN-infected pine, while symbiotrophs were more abundant in healthy trees [58]. All these HTS studies have already shed insight on fungal communities’ structure caused by PWN infection. Surprisingly, there are few references to the diversity and abundance of ophiostomatoid fungal communities frequently isolated from PWN-infected trees (see next section).

### 3.2. Close Relationships between Blue-Stain Fungi and the PWN

Pine trees infected with PWNs are often infested with bark beetles (Curculionidae) carrying a wide range of ophiostomatoid fungi, also commonly called blue-stain fungi, either in specialized structures (e.g., mycangia) or on the exoskeleton [59,60]. These bark beetles are limited to colonizing weak, or recently killed, trees, yet there are species capable of developing in living trees and even killing healthy trees [60]. The beetle-associated fungi belong to the unrelated orders Ophiostomatales (*Ophiostoma* s. l., *Grosmannia,* and *Ceratocystiopsis*) and Microscales (*Ceratocystis*) [61], and they are mostly necrotrophic pathogens of varying virulence that are able to colonize the phloem and xylem of pine species [62]. The reproduction structures of some of these blue-stain fungi were detected in the tunnels of *Monochamus* spp. as well as in the pine trees or nematode, thus suggesting an association with the disease [42,47,63] (Table 2). In the later stages of PWD, the PWN is able to switch from its plant-parasitic mode to a fungal-feeder depending on the available pine tree fungi for its nutrition [20]. Kobayahi et al. [40] showed that *Ceratocystis*, *Fusarium*, *Macrophoma,* and *Pestalotia* seemed to be a favourable food for the PWN. Fukushige [53] also reported that only *Ceratocystis* sp. was the most suitable fungi, among others, for PWN, leading to a quick increase in the population four weeks after inoculation in pine segments. To understand the factors influencing the number of PWNs carried by the insect-vector *M. alternatus*, Maehara and Futai [64,65] inoculated wood blocks containing the beetle (prior its emergence) with different species of fungi isolated from infected *P. thunbengii* and *P. densiflora*. The authors confirmed that the PWN was only able to grow densely and account for a higher number of nematodes transferred to the beetle, when in the presence of *O. minus*. Thus, they concluded that only the most prevalent species of fungi in the killed pine trees could help determine the number of PWNs carried by the beetles emerging from the wood. Later, the same authors analyzed the temporal changes in the PWN population and the percentage of dispersal juveniles (J_III_) of PWNs on *P. densiflora* branches segments [66], reporting that only *O. minus*, *Macrophoma* sp., and *Trichoderma* sp. 1 could heavily increase the PWN population and the number of J_III_ over 12 weeks after inoculation. In field surveys in Kyoto and Ibaraki Prefecture (Japan), Maehara et al. [52] examined the effect of blue-stain fungi on the number of PWNs carried by *M. alternatus* emerging from logs of pine wilt-killed of *P. densiflora*. The authors confirmed that blue-stain fungi could be isolated from 90% of wilted pine, and that the number of J_IV_ nematodes carried by the beetle was significantly affected by the species of blue-stain fungi (Table 2), individual pine trees, and wood water content [52]. Using sterilized branches of *P. thunbergii*, Wang et al. [67] co-inoculated axenic PWNs with different species of fungi isolated from healthy and wilted *P. densiflora* in order to study the relationship between the existence and distribution of fungi and the multiplication and distribution of PWNs. The multiplication of the nematode was only successful in sterilized branches inoculated with *Cryptosporiopsis* sp. and *Leptographium* sp. [67]. Niu et al. [68] compared the propagation rate of PWNs treated with a monoterpene ratio representative of blue-stain infected pines (*Sporothrix* sp.) (137.8 mg/mL of α-pinene:β-pinene in a ratio of 1:0.8) and monoterpene ratio of healthy pines, or pines damaged by *M. alternatus* feeding (137.6 mg/mL of α-pinene:β-pinene in a ratio of 1:0.1). From this, Niu et al. found that PWN growth was significantly higher in the blue-stain infected pine monoterpene ratio [68]. The authors suggested that the PWN uses high monoterpene concentrations and native blue-stain fungi to improve its propagation and overcome host resistance. Zhao et al. [69] demonstrated that *Sporothrix* sp. had a strong positive effect on the population and prevalence of the invasive PWN-native beetle symbiosis in the xylem of trees. The fragrant diacetone alcohol released from the wood infected by *Sporothrix* sp. promoted fecundity of the nematode and the growth and survival of the beetle [69]. Togashi et al. [70] also showed an increase in the PWN population resulting from the presence of *O. minus*, although the effect on *M. alternatus* larvae or the rate of development to adulthood was not observed due to experimental differences reported in other studies [71] (i.e., pine bolts, instead of an artificial diet, were provided to *M. alternatus*).

### 3.3. Naturally Occurring Fungal Communities for the Control of PWN

While exploring PWN growth and development by the fungi of non-infected and infected pine trees, several fungi species showed promising results in the control of nematode populations (Table 2). Naturally occurring Ascomycota fungi from the genera *Aureobasidium*, *Aspergillus*, *Cephalosporium*, *Fusarium*, *Gliocladium*, *Mucor*, *Mortierella*, *Penicillium*, *Rhizoctonia*, and some species of *Trichoderma* and *Verticillium,* affected PWN survivability in in vitro or in vivo bioassays [40,52,53,54,64,65]. To further study the effect of *Trichoderma* sp. on PWN suppression and transmission by *M. alternatus*, Maehara et al. [72] inoculated several isolates of *Trichoderma* spp. into wilt-killed *P. desinflora* logs. Beetles from logs treated with *Trichoderma* sp. 3 carried less than 1000 nematodes. The authors suggested that combining the use of this fungus for PWN control with the entomopathogenic fungus *Beauveria bassiana* for *M. alternatus* control could represent a potential biocontrol application in PWD [72]. The entomopathogenic fungus *B. pseudobassiana*, isolated from naturally infected *M. galloprovincialis* in Spain, has also showed potential as a natural insect population regulator [83], and it may also be feasible in combination with other PWN control agents.

Nematophagous fungi have evolved different strategies to attack nematodes (free living or plant parasitic), among which the most interesting for biological control applications are the nematode-trapping fungi and the endoparasitic fungi [84]. The nematode-trapping fungi can produce specialized adhesive hyphal networks, knobs, constricting rings, and hydrolytic enzymes to trap and penetrate nematodes [85]. An extracellular serine protease (Ac1) of the nematode-trapping fungi *Arthrobotrys conoides* has been successfully tested on PWNs [73]. Ac1 was found to be effective at immobilizing the free-living *Panagrellus redivivus* and PWNs [73]. Previously used for the control of other plant-parasitic nematodes [86,87,88], the nematode-trapping fungi *Drechslerella dactyloides* (isolates CNU09125 and CNU091026) showed high efficiency, trap-forming, and capture ability against PWNs, trapping 100% of juveniles within 24 h after inoculation [74]. The nematophagous fungi *Verticillum* sp. Was the first recorded endoparasite isolated from the PWN [53]. Later, Liou et al. [75] isolated and characterized *Esteya vermicola* as a potential biocontrol agent of PWNs. In in vitro assays, PWN populations could be completely killed by *E. vermicola* in 8 to 10 days [75]. In 2001, *E. vermicola* was patented in the USA for PWN control [89]. New strains of *E. vermicola* CNU120806 were isolated in Korea [76] and tested together with the nematode-trapping fungus *A. brochopaga,* and the nematode-feeding fungus *B. cinerea* for nematode attraction to living mycelia and exudative substances [77]. The PWNs showed the strongest attraction to *E. vermicola* CNU120806 avolatile exudative and volatile organic compounds (VOCs) [77]. Lin et al. [78] demonstrated that VOCs from *E. vermicola* living mycelia could mimic volatiles from the host pine tree by producing monoterpenes α- and β-pinene, and the terpenoid camphor, thus explaining PWN’s attraction to *E. vermicola*. More detailed discussion on the advances of *E. vermicola* as a biocontrol agent of PWD can be found in Chu et al. [90] and Yin et al. [91]. More recently, a novel species of *Esteya* was described, *E. floridanum* sp. nov. [79]. This species was recovered from the head of the ambrosia beetle *Myoplatypus flavicornis* (Curculionidae: Platypodinae) in *P. taeda* and showed a similar infection process as *E. vermicola* with a high infectivity rate towards PWNs [79].

The use of bioactive metabolites with nematicidal activities, extracted from naturally occurring fungi from different environments, is an unexplored approach to PWN control. Metabolites of the fresh-water fungus *Caryospora callicarpa* YMF1.01026 (namely tetradecalactone metabolites caryospomycins A to C) exhibited moderate killing activity against PWNs [80]. Three compounds isolated from the endophytic fungi *Geotrichum* sp. AL4, found on leaves of *Azadirachta indica,* showed noticeable activities against the nematode [81]. Culture filtrates of *Acremonium* sp. BH0531, obtained from seawater, exhibited the highest PWN mortality rate of (ca. 93% mortality rate) in in vitro trials [82].

## 4. Conclusions

Plant microbiome is considered a very propitious strategy for fostering plant protection against abiotic and biotic stressors [92]. Research on PWD-associated mycoflora has been slowly progressing since the first studies in the mid-seventies. HTS technologies have enriched the narrow vision of culturomic studies, and we now know that the presence of PWNs can affect the fungal diversity of the infected trees. Ophiostomatoid, or blue-stain fungi, often recovered from wilted pine trees, are considered the most determinative biotic factors for multiplication and distribution of PWNs inside the tree and in the insect-vector. Naturally occurring fungi, endophytic or nematophagous (e.g., nematode-trapping fungi *D. dactyloides* or the endoparasitic fungi *E. vermicola*), should be further explored as new tools for PWD management.

## Figures and Tables

**Figure 1 jof-07-00780-f001:**
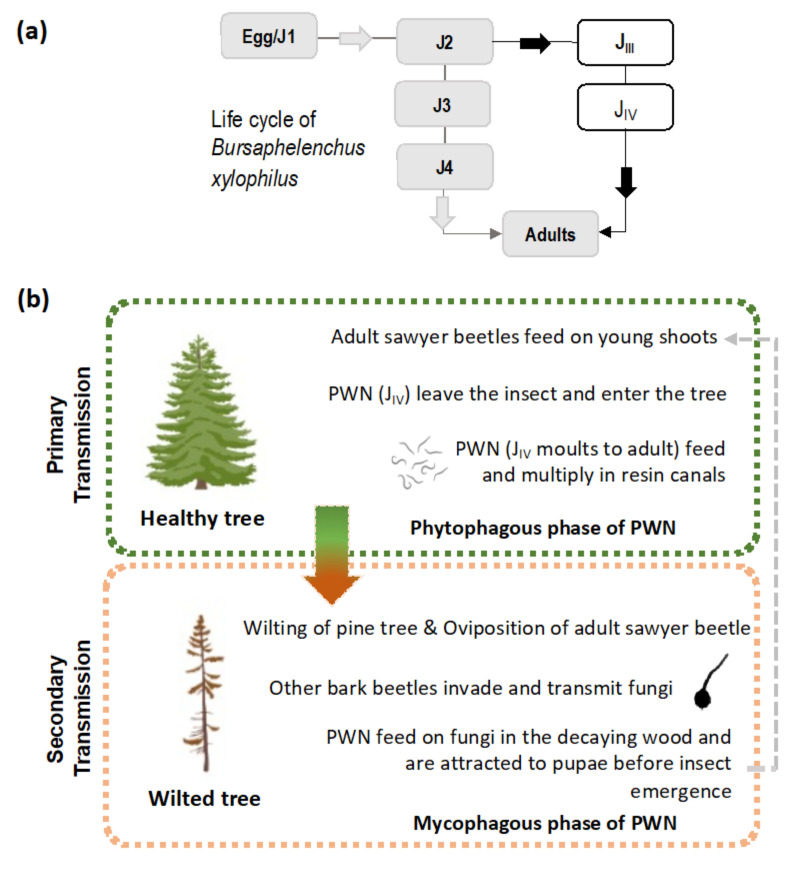
Pine Wilt Disease pathosystem. (**a**) Life cycle of the pinewood nematode *Bursaphelenchus xylophilus* (grey arrows indicate propagative stage; black arrows indicate dispersal stage); (**b**) Primary and secondary transmission of *B. xylophilus* by the sawyer beetle *Monochamus* sp. (grey arrow indicates interconnection between transmission modes). Image of a healthy and wilted tree, retrieved from Biorender^®^.

**Figure 2 jof-07-00780-f002:**
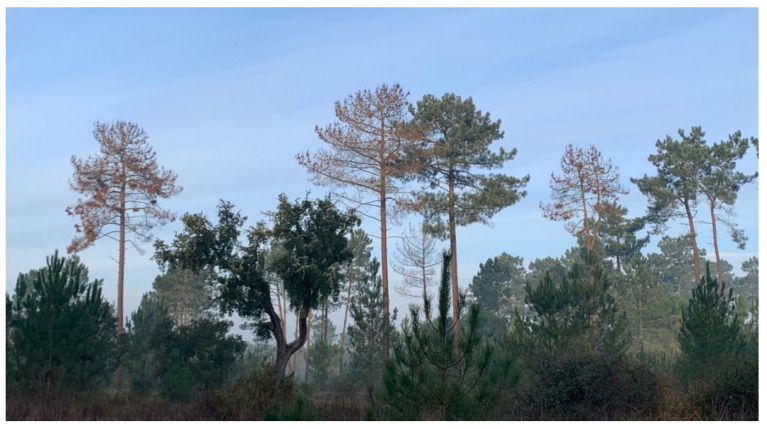
Symptoms of Pine Wilt Disease—wilted *Pinus pinaster* trees (pine trees with brown canopy) surrounded by assymptomatic *P. pinaster* (greenish canopy without PWD manifestation). Location: Companhia das Lezírias (Portugal; 38°49′17.6″ N 8°52′20.5″ W), in January 2020.

**Table 2 jof-07-00780-t002:** Nematode-fungi interactions in pine wilt disease (PWD). List of fungal species with a positive effect on pine wood nematode (PWN) multiplication and distribution, and potential agents for biocontrol applications with respect to PWD management.

Fungi in Interaction with PWNs		Mode of Action	References
Growth promotors	*Botrytis cinerea*	Increase of PWN population growth in in vitro mycelial fungi	[53]
	*Ceratocystis* sp.	Increase of PWN population growth in in vitro mycelial fungi and in pine segments of *Pinus densiflora*	[39,40,53]
	*Diplodia* sp.	Increase of PWN population growth in in vitro mycelial fungi	[53]
	*Pestalotia* sp.	Increase of PWN population growth in in vitro mycelial fungi and in pine segments of *Pinus thunbergii*	[39,40]
	*Macrophoma* sp.	Increase of PWN population growth in in vitro mycelial fungi and in pine segments *of Pinus thunbergii*	[39,40,66]
	*Fusarium* sp.	Increase of PWN population growth in in vitro mycelial fungi	[39,40]
	*Ophiostoma minus*	Increase of PWN population growth in wood blocks of *Pinus thunbergii;* Increase of PWN population growth in *Pinus densiflora* bolts; Increased no. of PWNs carried by emerging *Monochamus alternatus*	[64,65,66,67,70,71]
	*Leptographium* sp.	Increase of PWN population growth in logs of *Pinus densiflora;* Increase of axenic PWN population on autoclaved cuttings of *Pinus thunbergii*	[52,53,67]
	*Crytosporiopsis* sp.	Increase of PWN population growth in wood blocks of *Pinus densiflora*	[52]
	*Sporothrix* sp.	Increase of PWN population growth in in vitro mycelial fungi	[68,71]
	*Ophiostoma ips*	Increase of PWN population growth in in vitro mycelial fungi and in segments of *Pinus thunbergii*	[68,69,70]
	*Leptographium pine-densiflorae*	Increase of PWN population growth in in vitro mycelial fungi	[71]
	*Trichoderma* sp. 1	Increase of PWN population growth in in vitro mycelial fungi	[52]
	*Trichaptum abietinum* *Arthrobotrys* sp. *Gloeophyllum striatum* *Cryptoporus volvatus*	Increase of PWN population growth in in *Pinus densiflora;* Increased no. of PWNs carried by emerging *Monochamus alternatus*	[66]
Potential for biocontrol applications	*Alternaria* sp. *Epicoccum* sp.	Decrease of PWN population growth in in vitro mycelial fungi	[53]
	*Aureobasidium* sp. *Aspergillus* sp. *Gliocladium* sp. *Mucor* sp. *Mortierella* sp. *Penicillium* sp. *Rhizoctonia* sp.	Decrease of PWN population growth in in vitro mycelial fungi	[54]
	*Cystidiophorus castaneus*	Decrease of PWN population growth in *Pinus densiflora;* Decrease the no. of PWNs carried by emerging *Monochamus alternatus*	[66]
	*Cephalosporium*	Decrease of PWN population growth in in vitro mycelial fungi	[53,54]
	*Fusarium* sp.	Decrease of PWN population growth in in vitro mycelial fungi	[53,54]
	*Pycnoporus coccineus*	Decrease of PWN population growth in *Pinus densiflora;* Decrease in the no. of PWNs carried by emerging *Monochamus alternatus*	[66]
	*Trichoderma* sp.	Decrease of PWN population growth in in vitro mycelial fungi and in wood blocks of *Pinus densiflora* and *P. thunbergii;* Decrease in the no. of PWNs carried by emerging *Monochamus alternatus*	[53,64,65,66,72]
	*Verticillium* sp.	Endoparasitic fungi; Decrease of PWN population growth in in vitro mycelial fungi and in wood blocks of *Pinus densiflora* and *P. thunbergii;* Decrease in the no. of PWNs carried by emerging *Monochamus alternatus*	[65,66]
	*Arthrobotrys conoides*	Nematode-trapping fungi; Extracellular enzyme Ac1 with nematostatic effect on PWN	[73]
	*Drechslerlia dactyloides*	Nematode-trapping fungi	[74]
	*Esteya vermicola*	Endoparasitic fungi; Decrease of PWN population growth in in vitro feeding trials; Volatile compounds attractive to PWN	[74,75,76,77,78]
	*Esteya floridanum*	Endoparasitic fungi; Decrease of PWN population growth in in vitro feeding trials	[79]
	*Caryospora callicarpa*	Nematicidal activity of caryospomycins A to C metabolites exhibit moderate killing of PWN	[80]
	*Geotrichum* sp. AL4	Nematicidal activity against PWN	[81]
	*Acremonium* sp. BH0531	Nematicidal activity against PWN	[82]

## Data Availability

Not applicable.

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
