# Peer review of "Insights into the Role of Fungi in Pine Wilt Disease"

_jof, 2021, doi:10.3390/jof7090780_

Round 1

Reviewer 1 Report

In the review by Vicente et al., the authors describe the current state of knowledge of fungal communities associated with several Pinus species when affected by pine wilt disease. The authors provide a table that reviews the taxa found through culture-dependent studies and also describe diversity data from the few studies that have been conducted with high throughput sequencing. I thought that the manuscript overall was well done, but I do have a few comments. As a microbial ecologist, my comments on this manuscript focus on the fungal community data that are reviewed and presented.

Major comments:

  1. In the title and on line 26 of the abstract, the authors describe their review as documenting “interactions” between the PWN and fungi. I feel that this is not an accurate characterization of the review, since all of section 2 describes the organisms involved in PWD and section 3.1 discusses fungal communities/diversity associated with diseased and healthy trees. Sections 3.2 and 3.3 describe interactions between the PWN and fungi. But, this is less than half of the manuscript. In addition, the use of the term “mycobiota” in the manuscript title and in the title for section 3 (line 112) doesn’t seem correct to me. To me, this term is used to describe the fungi of a place or a habitat (e.g., the mycobiota of oak leaves). I’ve not heard of the mycobiota of a disease (in this case “the mycobiota of pine wilt disease”). To me, it is more accurate to discuss the mycobiota of pine trees that experience PWD symptoms or the foliar mycobiota in the presence of PWN. Given this, I suggest changing the title, end of the abstract, and the use of the word “mycobiota” throughout.
  2. Throughout section 3, the authors refer to the functional guilds of several genera, such as saprotrophs being isolated from trees affected by PWD. How were these taxa defined as saprotrophic? There are programs available, such as FUNGuild (https://doi.org/10.1016/j.funeco.2015.06.006) and FungalTraits (https://doi.org/10.1007/s13225-020-00466-2) that can use the taxonomic identity of fungi to predict functional guild. Did the authors confirm the saprotrophic guild assignment using either of these? In addition, it would be helpful for the reader if the blue-stain taxa were highlighted in some fashion in Table 2. (The names could be in bold or a footnote could be added.)
  3. Throughout the manuscript and tables, the authors need to pay attention to the words that are italicized. All genus names need to be italicized (see section 3.1 especially). In addition, the manuscript has several words that are italicized that do not need to be (especially in Table 2).

Minor Comments

Line 24 – PWN is not defined before this first use in the abstract. Please change to “pinewood nematode”.

Line 52-53 – I’m not sure what is meant by “intervenient” here. Do you mean “organisms”?

Line 65 – Add ending paranthesis.

Lines 65-70 – The description of the PWN life cycle is a little confusing here for me; especially the switch to Roman numerals (e.g., J3 to JIII). As stated above, I am a microbial ecologist, so that may be why this was a little confusing to me. Perhaps a figure, though, of the nematode life cycle or some more details here could help readers.

Liens 79-80 – I’m a little confused here about secondary transmission. Is it secondary because the young beetles will leave the dead/decaying tree to feed on a young pine branch? Again, I think a figure of the nematode life cycle could help.

Line 118 – This should be “community”, not “communities.

Line 120 – I would add “and composition” to the section title, as both diversity and taxonomic composition are reviewed.

Lines 124-127 – This statement here, to me, ignores decades of DNA profiling techniques, such as RFLP, TRFLP, and DGGE. These are molecular methods and have been used in combination with fungal morphological identification. It also feels out of place since these statements are not specific to studies of PWD. My suggestion is to omit this statement entirely, as to me, it seems to omit many years of molecular work.

Line 164 – This should be “genera”

Line 166 – Is there “personal communication” necessary with the conference presentation reference?

Line 189-191 – This statement is a bit out of place here, to me, because bacteria are profiled with the 16S region. Perhaps specify this in some way here.

Line 203 – This should be “rhizosphere”, not “rhisozpheric.”

Lines 202-209 – The description of the Wei et al. results were confusing. At first it seems that rhizosphere fungal communities were affected by disease status, but then it seemed like they were not. Please be clear about whether rhizosphere diversity was altered by PWD in this study. Also please clarify which plant compartments were studied overall (leaf, branch, stem, root, and rhizosphere are all mentioned at some point). Listing these outright and how fungal communities in these compartments changed with disease would be helpful.

Lines 214-216 – Please clarify that the abundances of these genera were higher on wilted than healthy pine trees.

Line 228 – It’s more accurate to say “community structure” and this should not be italicized.

Lines 228-230 – I think this is meant as a transition to the next section, so perhaps add a parenthetical statement “(see next section)”. Otherwise, as a reader, I wonder why there are no references to the few studies of ophiostomatoid fungi and PWD.

Line 272 – This should be “where” not “were”.

Lines 333-334 – Based on the reviewed literature in section 3.1, there are contrasting reports of how PWD affects fungal diversity, and this seems to be based on plant compartment. I would soften this conclusion here, or be more clear in section 3.1 that the high throughput sequencing studies to date show reduced fungal diversity in needles so that this statement in the conclusion is well-supported.

Author Response

The authors are thankful for your careful revision and for all the comments and corrections to our manuscript. To ease your revision, we have provided the line number for each modification in reference to the new version of the manuscript.

Reviewer 2 Report

This review manuscript is focused on interactions between fungi and the other main elements of pine wilt disease, and contains important knowledge. However, the manuscript has some weaknesses, especially in Section 2 and Tables 1-2. The weaknesses are as follows.

1. The most contents of Section 2 (The complexity of PWD) are general explanations of the interactions among PWN, Monochamus beetles and pine trees, and are not directly related to the main subject of this review.

2. Table 1 has several misunderstandings about the contents (fungi) of references, e.g., Ref. 39. Please check them again. There is not an explanation of “Pp.”

3. L161-162: Ref. 53 is not “This study.”

4. L239-242: Were the reproduction structures of blue-stain fungi detected in the nematode? Please add the references.

5: Table 2 has some errors in regard to “Experimental Trial” of references. Please check them again.

6. L247-249: Ref 39 is not “Kobayashi et al.”

7. L271-273: The meaning of this sentence is unclear.

8. L297-300: Ref. 39 does not contain the contents of the entomopathogenic fungus Beauveria bassiana.

9. L314-316: Fukushige (1991) (Ref. 43) also isolated an endoparasite of PWN, Verticillium sp.

Minor changes

L254: P. thunbergii → P. densiflora and P. thunbergii

L255: O. ips → O. minus

L284: O. minor → O. minus

Author Response

(The authors gave the same response as above.)

Reviewer 3 Report

All of scientific names should be  written by italic type. Authors should check them carefully, particularly in page 5.

Author Response

(The authors gave the same response as above.)
